# Consistency-based Black-box Uncertainty Quantification for Text-to-SQL by Similarity Aggregation

## Abstract

When does a large language model (LLM) know what it does not know? Uncertainty quantification (UQ) provides an estimate of the confidence in an LLM's generated output and is therefore increasingly recognized as a crucial component of trusted AI systems. UQ is particularly important for complex generative tasks such as *text-to-SQL*, where an LLM helps users gain insights about data stored in noisy and large databases by translating their natural language queries to structured query language (SQL). *Black-box* UQ methods do not require access to internal model information from the generating LLM, and therefore have numerous real-world advantages, such as robustness to system changes, adaptability to choice of LLM (including those with commercialized APIs), reduced costs, and substantial computational tractability. In this paper, we investigate the effectiveness of black-box UQ techniques for text-to-SQL, where the consistency between a generated output and other sampled generations is used as a proxy for estimating its confidence. We propose a high-level non-verbalized *similarity aggregation* approach that is suitable for complex generative tasks, including specific techniques that train confidence estimation models using small training sets. Through an extensive empirical study over various text-to-SQL datasets and models, we provide recommendations for the choice of sampling technique and similarity metric. The experiments demonstrate that our proposed similarity aggregation techniques result in better calibrated confidence estimates as compared to the closest baselines, but also highlight how there is room for improvement on downstream tasks such as selective generation.

## 1 Introduction

The process of translating natural language queries into SQL queries is an important endeavor in natural language processing, particularly within enterprise contexts where users such as business analysts, data engineers, and data scientists constantly query databases for data management, data analysis, and operational decision-making. Although there have been numerous advances in leveraging large language models (LLMs) for such tasks (Shaw et al., 2021; Gao et al., 2024a; Tai et al., 2023; Pourreza & Rafiei, 2024; Fan et al., 2024; Maamari et al., 2024), deploying *text-to-SQL* systems in real-world enterprise scenarios poses multifaceted challenges. For instance, the complexity of user queries in enterprise databases, unclear and even cryptic schemas with heavily abbreviated and domain specific column names, and data quality issues such as missing values and other inconsistencies all lead to inaccuracies in generated SQL queries. These are practical issues beyond more basic ones arising from the ambiguities of natural language queries and the intricacies of SQL syntax.

*Uncertainty quantification* (UQ) approaches in machine learning provide insights into the reliability of model predictions, and can therefore be a critical component of a real-world text-to-SQL system given the aforementioned challenges and complexities. In this paper, we use the term UQ to refer to estimating the confidence of generations, in our case for the text-to-SQL task. Our primary goal is to obtain predicted probabilities that are well *calibrated*, as gauged by how closely they align with the empirical accuracy of the predictions (Murphy & Epstein, 1967; Dawid, 1982). Specifically, we investigate the effectiveness of *black-box* UQ techniques for text-to-SQL, assuming access only to the model being used (such as an LLM) without requiring other model information such as the weights or even the token log probabilities. Such techniques have numerous practical advantages,

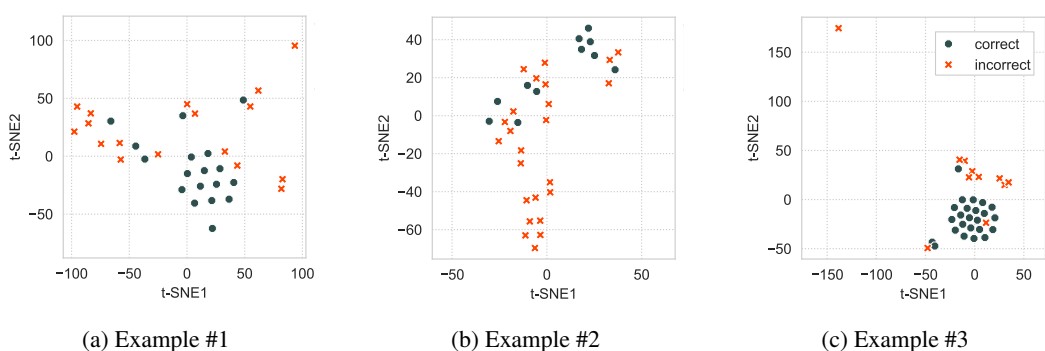

(a) Example #1      (b) Example #2      (c) Example #3

Figure 1: T-SNE projections of 35 generations each from a few-shot codellama model, for 3 instances from the Spider dev set. Correct/incorrect generations are labeled in blue/red respectively.

as they are robust to the constantly evolving landscape of LLMs and can easily adapt to system changes. Furthermore, they are usually computationally lightweight and can be quickly deployed at inference time. As a result, black-box UQ has become increasingly popular for tasks such as question answering, where they have demonstrated superior performance over baselines (Kuhn et al., 2022; Lin et al., 2024; Manakul et al., 2023; Cole et al., 2023). An open question remains however about how these methods perform in estimating confidence for more complex generative tasks, involving longer and more structured outputs such as SQL queries.

We pursue *consistency-based black-box UQ* for text-to-SQL, where the idea is to use the consistency between a generation and other sampled generations as a proxy for its confidence. The implicit underlying assumption behind consistency-based approaches is that when a generated response is more different from others, it is more likely to be incorrect, implying that responses that are consistently similar are more likely to be correct (Mitchell et al., 2022; Wang et al., 2023; Chen et al., 2024). Figure 1 visualizes three instances from the Spider dataset (Yu et al., 2018) for the text-to-SQL task, projecting semantic encodings of 30 generations for each instance while distinguishing between generations deemed to be correct vs. incorrect. We observe that correct generations tend to be closer to other generations, particularly other correct ones, and incorrect generations tend to be spread out more and lie on the border of or beyond correct generations.

Figure 2 outlines our proposed high-level procedure that aims at exploiting the afore-mentioned assumption for UQ. First, multiple outputs/samples are generated by the LLM through some sampling procedure. Pairwise similarities between samples can then be computed using any similarity metric of choice. Finally, these similarities are leveraged to provide confidence estimates for each generation of interest. Our methodological contributions are primarily in the third phase of Figure 2. In particular, rather than viewing the third phase as clustering outputs like in closely related work (Kuhn et al., 2022; Lin et al., 2024), we propose *similarity aggregation* as a framework for estimating confidences. In contrast to aggregating verbalized confidences (Xiong et al., 2024c), we aggregate pairwise similarities between generations, making our approach *non-verbalized* and therefore avoiding some empirically observed concerns around potential overconfidence when asking LLMs for probabilities (Hu & Levy, 2023; Xiong et al., 2024c). Verbalized confidence aggregation approaches are likely to struggle for semantic parsing tasks since answers are not unique and generated SQLs have syntactical requirements. Our consistency-based UQ methods typically provide higher confidence to generations that are more similar to others in aggregate, such as those in the interior of Figure 1 rather than near the boundary.

Note that in the illustrative example shown in Figure 2, the first two SQLs are correct but the third is incorrect; this is reasonably captured in this instance by the corresponding estimated confidence of 6%, which is lower than the estimates of 22% and 31% for the correct SQLs. The third generated SQL happens to be different from other generations, resulting in its low confidence estimate.

Our **contributions** and main **findings** are summarized as follows:

- We conduct an empirical investigation of consistency-based black-box UQ techniques for text-to-SQL using multiple benchmark datasets and LLMs. We are not aware of prior explorations of such techniques for generative outputs with structure and complexity such as SQL queries.

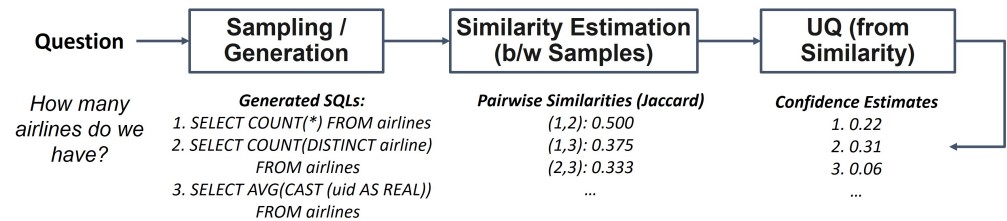

Figure 2: Procedure for black-box UQ that yields confidence estimates for various SQL generations from an LLM, based on a natural language question over a relational database. An illustrative natural language query from the Spider dataset is shown, along with example outputs at each step of the pipeline. 3 generated SQLs are shown out of 30 that were generated from a fine-tuned Deepseek model. The Jaccard metric is chosen as the similarity metric, and confidences are estimated using a proposed 'aggregation by classification' method with random forests (described later).

- We propose a high-level similarity aggregation framework for estimating confidences, including specific data-driven instances from this framework such as Bayesian aggregation and aggregation by classification. These require simple models learned from a small training set.
- We highlight several insights about consistency-based black-box UQ for text-to-SQL, for instance: sampling by varying the temperature can be beneficial for improving UQ performance, and that text/token-based similarity metrics such as Jaccard and Rouge-L are suitable choices of similarity metric. The proposed similarity aggregation techniques result in well-calibrated confidence estimates as gauged by our choice of calibration error evaluation metric, but there is room for improvement when using the confidence estimates in downstream applications, such as for deciding when to abstain from generation.

## 2 RELATED WORK

We briefly highlight relevant related work covering the evolution of text-to-SQL and UQ approaches.

### 2.1 TEXT-TO-SQL

Early work on mapping natural language utterances to a form of structured language for querying databases has focused on semantic parsing using either logic programs (Zelle & Mooney, 1996), λ-calculus (Zettlemoyer & Collins, 2005), or λ-DCS (Liang et al., 2013) as the underlying meaning representation. Other prominent work in natural language interfaces for databases includes PRE-CISE (Popescu et al., 2003), which translates questions to SQL queries and identifies questions that it is not confident about, and approaches that first generate candidate queries from a grammar and then rank them using tree kernels (Giordani & Moschitti, 2012). These methods rely on high quality grammars and are not suitable for tasks that require generalization to new schemas.

The next phase of text-to-SQL systems considered neural sequence-to-sequence models (Iyer et al., 2017; Zhong et al., 2017), often trained using a popular cross-domain text-to-SQL benchmark called Spider (Yu et al., 2018) in order to be schema independent. More modern text-to-SQL translation systems rely on pre-trained LLMs and either using various forms of constrained decoding (Scholak et al., 2021; Shaw et al., 2021), fine-tuning (Gao et al., 2024a), or various prompting techniques around decomposition and chain-of-thought (Tai et al., 2023; Li et al., 2024b; Pourreza & Rafiei, 2024; Talaei et al., 2024; Mandamadiotis et al., 2024). Several recent methods including top contributors to the Spider and BIRD (Li et al., 2024a) leaderboards demonstrate that fine-tuned open models can achieve sufficient performance (Gao et al., 2024a; Talaei et al., 2024; Xiong et al., 2024a).

There is also a growing body of literature on error detection and correction for text-to-SQL (Chen et al., 2023a;b; Lee et al., 2024), as well as the use of self-consistency and related approaches for voting among candidate generations (Gao et al., 2024a; Talaei et al., 2024). Several error detection and concept shift detection works demonstrate the applicability of UQ for their purposes (Vazhentsev et al., 2023a; Fadeeva et al., 2023). Similar attempts to use UQ methods for error parsing in text-to-SQL show good error coverage but low precision (Zeng et al., 2020; Li et al., 2020). These efforts are

primarily components intended to improve the execution accuracy of a text-to-SQL system and do not estimate confidences of SQL output. In contrast, here we are interested in developing a black-box UQ module that is able to provide well-calibrated confidence estimates, with at most a limited amount of data, for any state-of-the-art text-to-SQL system and potentially applicable to other generative tasks.

## 2.2 Uncertainty Quantification

Procedures for UQ typically estimate measures such as the variability or confidence of the LLM output. These methods are either *white-box* or *black-box*, where the former category operates on the premise that the internal state of the LLM including the model weights, logits, and/or embeddings are accessible. In contrast, *black-box* methods assume that all parameters during inference are unknown, allowing access only to the generations. In this case, an output's confidence is inferred by other means, such as by gauging the consistency of the output after paraphrasing the input prompt. An orthogonal categorization is whether a method is *verbalized* or *non-verbalized*, where the former category involves prompting an LLM to express uncertainty in natural language. This involves discerning different levels of uncertainty, such as through qualifying phrases (e.g. "I don't know" or "most probably"), verbalized words (e.g. "low" or "high"), or numbers (e.g. 50% or 90%).

**White-box Methods.** Common approaches to estimating an LLM's confidence include considering the minimum or average token-level probabilities (logits) or entropy (Huang et al., 2023; Vazhentsev et al., 2023b) coupled with a normalization mechanism to ensure consistency over outputs of different lengths (Murray & Chiang, 2018). Linguistic semantics such as token-level or sentence-level relevance can also be incorporated into these schemes to yield more effective confidence estimators (Duan et al., 2024). Kuhn et al. (2022) propose semantic entropy based clustering on multiple samples generated from the model and then estimating confidence estimates by summing the token-level probabilities in each cluster. Kadavath et al. (2022) suggest a verbalized method where the LLM first generates responses and then evaluates them as either True or False; the probability the model assigns to the generated token (True or False) determines the confidence level.

Other approaches consider the LLM's internal state such as embeddings and activation spaces. For instance, Ren et al. (2023) compute embeddings for both inputs and outputs in the training data, fit them to a Gaussian distribution, and estimate the model's confidence by computing the distance of the evaluated data pair from this Gaussian distribution. Some methods probe the model's attention layers to discriminate between correct and incorrect answers (Kadavath et al., 2022; Burns et al., 2023; Li et al., 2023; Azaria & Mitchell, 2023). Although these methods provide insights into the model's linguistic understanding, they typically require supervised training on specially annotated data.

**Black-box Methods.** One strand of research considers verbalized black-box methods, such as using an LLM to evaluate the correctness of its own generated answers in a conversational agent scenario (Mielke et al., 2022). Xiong et al. (2024c) conduct an empirical study on UQ for reasoning tasks, showing that LLMs tend to be overconfident when verbalizing their own confidence in the correctness of the generated answers and align poorly with the likelihood of factual correctness, which may pose significant safety risks in real-world deployments of LLMs. Other related work includes that of Lin et al. (2022) around fine-tuning GPT-3 to verbalize the uncertainty associated with the generated answers. Analysis in Hu & Levy (2023) reveals that LLMs' meta-linguistic judgments are less reliable than quantities derived directly from the model's token-level probabilities.

Another promising direction of work assumes that a model's lack of confidence correlates with various responses, often leading to hallucinatory outputs. In this case, confidence is typically estimated by analysing the consistency among various responses of the model. Specifically, Manakul et al. (2023) propose a simple sampling-based approach that uses consistency among generations to find potential hallucinations. Lin et al. (2024) calculate the similarity matrix between generations and then estimate the uncertainty based on the analysis of the similarity matrix, such as the sum of the eigen-values of the graph Laplacian , the degree matrix, and the eccentricity. Recent methods have also explored combining white-box and black-box approaches (Chen & Mueller, 2024; Shrivastava et al., 2023).

Our proposed approach falls within the black-box UQ category and relies on evaluating consistency among text-to-SQL generations. Although there is some prior work that incorporates data and model uncertainty for representation learning in parsing SQL (Qin et al., 2022) and other work that recognizes the challenges of using model logits for text-to-SQL (Stengel-Eskin & Van Durme, 2023), we are not aware of prior explorations of UQ for text-to-SQL that only rely on model API access.

## 3 METHODOLOGY

Suppose an LLM generates output $y$ for some input $x$. We assume there is an associated ground truth output $y^*$ for input $x$ as well as a binary reward $r \in \{0, 1\}$ from a reward function $r(x, y, y^*)$. We use *execution accuracy* as the performance measure for the text-to-SQL task considered here, where reward $r = 1$ if the generated and ground truth queries return the same result upon query execution on the underlying database. For other tasks such as open-ended QA, the reward could be 1 based on whether a text similarity metric (e.g. ROUGE) between the ground truth and generated output exceeds some predetermined threshold (Kuhn et al., 2022; Lin et al., 2024).

In this work, we propose an overarching framework with specific techniques that provide confidence estimates, possibly using a limited amount of training data. We denote the confidence of a generation $y$ for input $x$ as $c(x, y)$ and interpret it as the probability that it is correct, i.e. $c(x, y) = P(r(x, y, y^*) = 1) = P(y \in Y^*(x))$, where $Y^*(x)$ is the set of responses with reward 1. In the remainder of this section, we describe the components of the workflow depicted in Figure 2 for estimating $c(x, y)$.

### 3.1 SAMPLING

Consistency-based approaches begin with the generation of multiple samples/generations $y_1, \cdots, y_m$ for an input $x$. We explore three potential ways to generate samples:

- *Standard sampling* is when tokens are generated by sampling from the next-token probability distribution of the LLM at a fixed temperature, therefore enabling variable generations.
- *Temperature sampling* occurs when each sample is generated from a different temperature, thereby potentially further increasing the variability of generations.
- *Hybrid sampling* is a combination of the above methods, where multiple samples are generated from multiple temperatures.

While there are other means of generating diverse samples (Gao et al., 2024b), including approaches in text-to-SQL systems that modify the prompts (Bhaskar et al., 2023; Lee et al., 2024), our focus is primarily on leveraging temperature (Zhu et al., 2024); this provides sufficient variability, which is needed by consistency-based methods to help distinguish correct from incorrect responses in complex generations such as SQL. In practical usage, one may be interested in confidence estimates for only a subset of the generations, such as the ones most likely to be correct. Generations at higher temperatures could therefore potentially be used merely for obtaining better confidence estimates for those at lower temperatures.

### 3.2 COMPUTING PAIRWISE SIMILARITIES

After generating samples, consistency-based approaches rely on access to a similarity metric with which one can compute pairwise similarities $s(y_i, y_j)$ for all sample pairs. For our experiments, we consider two types of similarity metrics, all lying in the interval $[0, 1]$:

- *Token/text metrics*: We consider metrics that treat the samples as general text or sets of tokens, such as the Jaccard coefficient, variations of ROUGE metrics such as Rouge-1 and Rouge-L, and the cosine similarity between sentence BERT (sbert) (Reimers & Gurevych, 2019) representations of the generations.
- *SQL metrics*: We also consider similarity metrics specific to SQL queries, such as the binary metric of whether two generations belong to the same SQL output type among 3 categories (simple/join/nested) (Pourreza & Rafiei, 2024), as well as those that rely on parsing the SQL and comparing the contents of various clauses – Aligon (Aligon et al., 2014), Aouiche (Aouiche et al., 2006), and Makiyama (Makiyama et al., 2015). Makiyama has been shown to perform well among these on a query clustering task (Tang et al., 2022).

### 3.3 SIMILARITY AGGREGATION

The final phase of the pipeline relies on leveraging pairwise similarities for UQ. Recall that the underlying assumption behind consistency-based approaches is that correct generations are more similar to other generations than incorrect ones. Computing an aggregated similarity between a particular generation and other generations therefore acts as a proxy for correctness.

We present a simple yet broad perspective on consistency-based approaches that is applicable to any generative task. Rather than clustering generations such as around semantic equivalence (Kuhn et al., 2022), the confidence for sample $y_i$ can be estimated using a suitable aggregation function, $c_i = f(s_1, \cdots, s_m)$, where $s_k$ is the similarity between samples $y_i$ and $y_k$. A deterministic function $f(\cdot)$ implies that identical generations yield identical confidences, for the same sample set, which in our view is a desirable property.

The choice of aggregation function should reflect the underlying hypothesis around consistency-based methods, which is that more consistency is expected for correct answers. We highlight and propose the following 3 categories for choosing aggregation function $f(\cdot)$.

**Simple Aggregation.** A simple approach is to find an aggregate distance between $y_i$ and other generations, $\bar{d} = g(d_1, \cdots, d_m)$ where distance $d_k = 1 - s_k$, and compute $c_i = 1 - \bar{d}$ since the aggregate distance lies in $[0, 1]$. The rationale is that the consistency hypothesis suggests that a generation further removed from others is more likely to be incorrect. While any form of aggregation $g(\cdot)$ is possible, we use the arithmetic mean for experiments, for which the estimation simplifies to $c_i = \frac{1}{m} \sum_k s_k$. This form of aggregation is mathematically equivalent to the spectral clustering by degree approach in Lin et al. (2024) and therefore treated as a baseline for most experiments. We show later that this performs reasonably well on some (but not all) UQ metrics.

**Bayesian Aggregation.** We propose a Bayesian form of aggregation that updates beliefs about confidence using similarities. Specifically, we compute the posterior probability of generation $y_i$ being correct, given the evidence from similarities with respect to other generations:

$$P(y_i \in Y_i^* | s_1, .., s_{i-1}, s_{i+1}, .., s_m) = \frac{p_0 \prod_{k \neq i} P(s_k | y_i \in Y_i^*)}{p_0 \prod_{k \neq i} P(s_k | y_i \in Y_i^*) + (1 - p_0) \prod_{k \neq i} P(s_k | y_i \notin Y_i^*)},$$

where prior $p_0 = P(y_i \in Y_i^*)$. The formula makes two important assumptions: 1) similarity $s_k$ depends only on whether $y_i$ is correct, and 2) similarities $\{s_k\}_{k \neq i}$ are conditionally independent. The first assumptions reflects the consistency hypothesis since we may expect a less variable distribution if $y_i$ is correct, but the second assumption is made purely for simplicity and tractability.

Note that this approach requires a small training set to learn the parameters of the probabilistic model. For experiments, we assume Beta distributions for the conditional similarity distributions; this requires 5 parameters to be learned – prior $p_0$ and 2 parameters each for the 2 Beta distributions.

**Aggregation by Classification.** We also propose treating similarity aggregation as a classification task; specifically, we train a probabilistic classifier for whether a response is correct using supervised learning where similarities are features: $c_i = P(y_i \in Y_i^*) = f(s_1, .., s_{i-1}, s_{i+1}, .., s_m)$. This is a natural extension of simple aggregation where the function is learned using a small training set. Both this approach and the Bayesian approach are more likely to be effective when the sampling procedure for training is similar to that during test time. In practical applications such as text-to-SQL, this is straightforward to control and the training dataset can be easily compiled using a small labeled dataset with ground truth responses. We experiment with logistic regression and random forests as the probabilistic classifier but other methods are also applicable.

# 4 EXPERIMENTAL SETUP

We describe our experimental setup around choice of datasets, models, and evaluation metrics.

**Datasets.** We consider the following real-world text-to-SQL datasets:

- **Spider** (Yu et al., 2018) is a popular text-to-SQL benchmark which requires models to generalize to novel database schemas, and covers 138 domains with 200 different databases such as academic databases, booking systems, and geography-related databases. The dev set has 1034 queries.
- **Spider-Realistic** (Deng et al., 2021) is considered a more challenging version of the Spider dev set as it modifies the natural language queries in Spider in an attempt to reflect realistic scenarios where questions do not make explicit mention of column names. It comprises a total of 508 queries.
- **BIRD** (BIg Bench for LaRge-scale Database Grounded Text-to-SQL Evaluation) (Li et al., 2024a) is a recent cross-domain benchmark of 95 databases (33.4 GB), covering more than 37 professional domains, such as blockchain, hockey, healthcare, and education. The dev set includes 1533 queries.

**Models.** We consider the following LLMs for text-to-SQL:

- **Few-shot Codellama**: A 34B codellama instruct model (Rozière et al., 2024) which is a code-specialized version of Llama 2 trained with 500B tokens of code and code-related data.
- **Few-shot Granite**: A 34B Granite code instruct model (Mishra et al., 2024) trained on 3-4 trillion tokens sourced from 116 programming languages.
- **LoRA fine-tuned Deepseek**: A 33B Deepseek coder instruct model (Guo et al., 2024) trained on 2 trillion tokens from 80 programming languages, which is further fine-tuned with LoRA (Hu et al., 2022) for the text-to-SQL task using the training set of Spider.

Our chosen models are representative of those commonly deployed for text-to-SQL, with and without fine-tuning, and exhibit varying degrees of performance. We restrict ourselves to open-source models and therefore do not consider models from OpenAI. Interested readers may peruse the leaderboard for the BIRD[1] dataset to explore how various models and approaches compare on a recent text-to-SQL benchmark. Our UQ experiments can be conducted on a stand alone CPU machine, but we use GPU machines (typically NVIDIA A100s with more than 40GB memory) for generating samples from the various LLMs. Please see Appendix A for further experimental details.

**Evaluation Metrics.** Our main objective is to estimate **well-calibrated confidences for SQL generations**, i.e. probabilities that match empirical observations. Calibration help in enabling trust in a system's generations, for the benefit of both system builders as well as end users. There is significant discussion about the limitations of various calibration metrics in the literature (Nixon et al., 2019; Xiong et al., 2024b); in this work, we choose *adaptive calibrated error* (ACE), which bins confidence estimates into probability ranges such that each bin contains the same number of data points (Nixon et al., 2019). Formally, $ACE = \frac{1}{KB} \sum_{k=1}^{K} \sum_{b=1}^{B} |acc(b,k) - c(b,k)|$, where $acc(b,k)$ and $c(b,k)$ are the accuracy and confidence of adaptive calibration bin $b$ for class label $k$. ACE is suitable in our application as generations are often highly similar or dissimilar, resulting in similar confidence estimates for various generations. Furthermore, confidences from some techniques are often skewed heavily towards either 0 or 1. We set the # of bins $B = 5$ for all experiments.

We also consider two other metrics that evaluate how the confidence estimates may be utilized. The *Area Under the Receiver Operating Characteristic* (AUROC) computes the area under the curve of the false positive rate vs. true positive rate when confidences are used as a probabilistic classifier for the correctness of generations. When confidences are used for selective generation (El-Yaniv et al., 2010; Kamath et al., 2020), i.e. for rejecting a fraction of the instances that we are the least confident about, then the *Area Under the Accuracy Rejection Curve* (AUARC) is a suitable metric, as it computes the area under the curve of the fraction of rejected instances vs. accuracy on the non-rejected instances (Nadeem et al., 2009).

## 5 EMPIRICAL INVESTIGATION

We conduct a detailed empirical investigation around UQ for text-to-SQL, describing various experiments along with our insights and associated recommendations.

### 5.1 EFFECT OF SAMPLING TECHNIQUE

We explore the effect of choice of sampling technique using generations from a few-shot codellama model on the Spider dev set. For this experiment, we generate 5 samples each over 6 temperatures ($\{0.25, 0.5, \cdots, 1.5\}$) and evaluate the impact of different samples on all 3 UQ metrics for a generation at the lowest temperature (in this case 0.25). For standard sampling, only other samples generated at the same temperature are considered, as opposed to one sample each from other temperatures for temperature sampling. For hybrid sampling, all other samples are included. The difference between the various techniques arises from the different samples over which the consistency-based methods apply. Our objective is to analyze which situation is most effective for UQ.

Figure 3 compares sampling techniques for black-box UQ with 3 selected similarity metrics – Jaccard, Rouge-L, and the SQL output type – using similarity aggregation by arithmetic mean. The results show benefits from temperature sampling and hybrid sampling over standard sampling; this

---

[1]`https://bird-bench.github.io/`

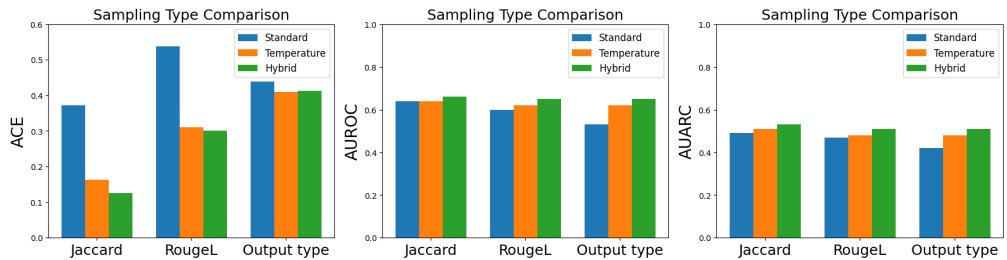

Figure 3: Comparing the effect of sampling technique for consistency-based black-box UQ using generations from a few-shot codellama model on the Spider dev set. Similarity aggregation was done using the arithmetic mean of pairwise similarities, using one of 3 similarity metrics: Jaccard, Rouge-L, and SQL output type. Lower ACE is better, whereas higher AUROC and AUARC are better. Error bars were computed over 5 splits of the Spider dev set but are not shown for readability.

Table 1: Comparing different similarity metrics and aggregation approaches for consistency-based black-box UQ using generations from a few-shot codellama model on the Spider Realistic dev set. We include 6 similarity metrics (across both SQL and token/text categories), 2 evaluation metrics (ACE and AUROC), and 5 UQ techniques – 2 baselines of spectral clustering with eccentricity and aggregation by arithmetic mean, and 3 proposed methods of Bayesian aggregation with conditional Beta distributions, and aggregation by classification using logistic regression and random forests. Error bars are from max. and min. values over 5 runs, each with a random $50\%$train / $50\%$test split.

| Eval. Metric | ACE ↓ | | | | | AUROC ↑ | | | | |
| | Baselines | | Proposed | | | Baselines | | Proposed | | |
| | spec-ecc | arith | beta | log-reg | rand-for | spec-ecc | arith | beta | log-reg | rand-for |
| --- | --- | --- | --- | --- | --- | --- | --- | --- | --- | --- |
| Makiyama | 0.516 ±0.014 | 0.155 ±0.014 | 0.208 ±0.104 | 0.091 ±0.008 | 0.095 ±0.012 | 0.27 ±0.01 | 0.73 ±0.02 | 0.74 ±0.01 | 0.72 ±0.02 | 0.73 ±0.01 |
| Output type | 0.384 ±0.009 | 0.316 ±0.022 | 0.306 ±0.053 | 0.128 ±0.011 | 0.126 ±0.013 | 0.37 ±0.03 | 0.63 ±0.03 | 0.63 ±0.02 | 0.63 ±0.03 | 0.63 ±0.02 |
| Jaccard | 0.515 ±0.016 | 0.080 ±0.007 | 0.146 ±0.028 | 0.084 ±0.007 | 0.053 ±0.008 | 0.27 ±0.02 | 0.77 ±0.02 | 0.77 ±0.03 | 0.76 ±0.03 | 0.77 ±0.02 |
| Rouge1 | 0.472 ±0.020 | 0.196 ±0.014 | 0.132 ±0.024 | 0.069 ±0.008 | 0.055 ±0.017 | 0.27 ±0.02 | 0.75 ±0.03 | 0.80 ±0.02 | 0.80 ±0.01 | 0.80 ±0.01 |
| Rouge-L | 0.491 ±0.020 | 0.180 ±0.013 | 0.136 ±0.024 | 0.070 ±0.007 | **0.050** ±0.008 | 0.25 ±0.02 | 0.77 ±0.03 | **0.81** ±0.01 | **0.81** ±0.01 | 0.80 ±0.01 |
| Sbert-cos | 0.442 ±0.009 | 0.354 ±0.012 | 0.128 ±0.028 | 0.071 ±0.012 | 0.054 ±0.010 | 0.37 ±0.02 | 0.69 ±0.02 | 0.79 ±0.01 | 0.78 ±0.01 | 0.79 ±0.01 |

is particularly notable for ACE where temperature sampling shows substantial performance gain for the Jaccard and Rouge-L metrics. Similar trends are observed for other aggregation methods, models, and datasets, indicating that the **variability of SQL generations across temperatures aids consistency-based black-box UQ methods, particularly for calibration metrics such as ACE**.

## 5.2 EFFECT OF SIMILARITY METRIC AND AGGREGATION TECHNIQUE

We investigate the choice of similarity metric and similarity aggregation technique using generations from a few-show codellama model on the Spider Realistic dataset. Temperature sampling over 6 temperatures is used for generations, and evaluations are performed using all 6 samples across all queries in the dataset. We split the data randomly into half for train/test sets, and repeat the experiment 5 times so as to study the variability of the results.

The rows in Table 1 correspond to 6 similarity metrics and the columns correspond to 5 UQ techniques with evaluations along 2 metrics – ACE and AUROC. For baselines, we consider two spectral clustering approaches for UQ that leverage a graph Laplacian matrix computed from pairwise similarities – one that uses eccentricity and another that uses degree (Lin et al., 2024); as mentioned previously, the latter is equivalent to simple aggregation using arithmetic mean.

Comparing similarity metrics, we see that Rouge-L performs the best for both evaluation metrics, although all token/text-based metrics perform well on ACE with a powerful aggregation method

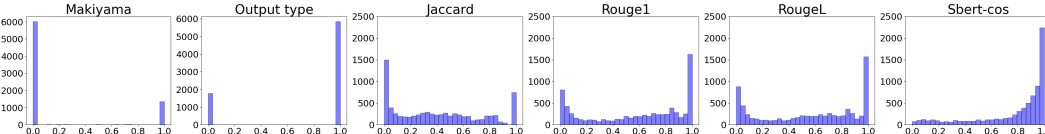

Figure 4: Histograms of pairwise similarities between generations from temperature sampling, using the few-shot codellama model on the Spider Realistic dataset for 6 similarity metrics.

Table 2: Comparing different UQ approaches on generations from 3 models on the Spider dev dataset. The non black-box (non BB) and black-box (BB) baselines are described in the main text. We consider 2 similarity metrics for the baseline and proposed black-box methods: Jaccard and Rouge-L.

| Model | | FS Codellama | | FS Granite | | FT Deepseek | |
|---|---|---|---|---|---|---|---|
| | | ACE ↓ | AUROC ↑ | ACE ↓ | AUROC ↑ | ACE ↓ | AUROC ↑ |
| Non BB (baselines) | always 0 | $0.132_{\pm 0.006}$ | $0.50_{\pm 0.00}$ | $0.083_{\pm 0.007}$ | $0.50_{\pm 0.00}$ | $0.164_{\pm 0.006}$ | $0.50_{\pm 0.00}$ |
| | always 1 | $0.368_{\pm 0.006}$ | $0.50_{\pm 0.00}$ | $0.417_{\pm 0.007}$ | $0.50_{\pm 0.00}$ | $0.336_{\pm 0.006}$ | $0.50_{\pm 0.00}$ |
| | avg. prob | $0.654_{\pm 0.012}$ | $0.53_{\pm 0.01}$ | $0.632_{\pm 0.012}$ | $0.65_{\pm 0.01}$ | $0.544_{\pm 0.012}$ | $0.52_{\pm 0.01}$ |
| | p(True) | $0.784_{\pm 0.005}$ | $0.52_{\pm 0.01}$ | $0.892_{\pm 0.003}$ | $0.63_{\pm 0.02}$ | $0.703_{\pm 0.009}$ | $0.47_{\pm 0.01}$ |
| BB (baselines) | spec-ecc; jaccard | $0.201_{\pm 0.006}$ | $0.37_{\pm 0.01}$ | $0.550_{\pm 0.009}$ | $0.20_{\pm 0.01}$ | $0.258_{\pm 0.012}$ | $0.36_{\pm 0.01}$ |
| | spec-ecc; rouge-L | $0.182_{\pm 0.004}$ | $0.37_{\pm 0.03}$ | $0.503_{\pm 0.007}$ | $0.21_{\pm 0.01}$ | $0.220_{\pm 0.010}$ | $0.36_{\pm 0.01}$ |
| | arith; jaccard | $0.238_{\pm 0.009}$ | $0.68_{\pm 0.01}$ | $0.070_{\pm 0.005}$ | $0.81_{\pm 0.01}$ | $0.088_{\pm 0.011}$ | $0.70_{\pm 0.01}$ |
| | arith; rouge-L | $0.418_{\pm 0.011}$ | $0.67_{\pm 0.01}$ | $0.142_{\pm 0.006}$ | $0.79_{\pm 0.01}$ | $0.242_{\pm 0.011}$ | $0.68_{\pm 0.01}$ |
| BB (proposed) | bayes-beta; jaccard | $0.298_{\pm 0.017}$ | $0.70_{\pm 0.02}$ | $0.317_{\pm 0.023}$ | $0.80_{\pm 0.01}$ | $0.326_{\pm 0.021}$ | $0.70_{\pm 0.01}$ |
| | bayes-beta; rouge-L | $0.382_{\pm 0.010}$ | $0.71_{\pm 0.01}$ | $0.338_{\pm 0.019}$ | $0.82_{\pm 0.01}$ | $0.369_{\pm 0.010}$ | $0.70_{\pm 0.01}$ |
| | clf-rf; jaccard | $\mathbf{0.045}_{\pm 0.008}$ | $0.71_{\pm 0.01}$ | $\mathbf{0.029}_{\pm 0.005}$ | $\mathbf{0.86}_{\pm 0.01}$ | $0.037_{\pm 0.007}$ | $\mathbf{0.71}_{\pm 0.01}$ |
| | clf-rf; rouge-L | $\mathbf{0.045}_{\pm 0.017}$ | $\mathbf{0.72}_{\pm 0.02}$ | $0.031_{\pm 0.010}$ | $\mathbf{0.86}_{\pm 0.01}$ | $\mathbf{0.034}_{\pm 0.006}$ | $0.70_{\pm 0.01}$ |

such as a random forest classifier. For instance, when our proposed methods are applied to the sentence BERT cosine similarity metric, performance is competitive with other metrics such as Jaccard and Rouge-L. **The proposed aggregation methods are better calibrated than baselines, as evaluated by ACE**. For AUROC, our proposed aggregation methods sometimes only provide marginal improvements over averaging similarities with the arithmetic mean baseline, showcasing that even **simple aggregation can be beneficial using a well-chosen similarity metric**. AUARC is not shown in these tables as trends are similar to those for AUROC. Results for the same experiment conducted on queries in the BIRD dev dataset are shown in Appendix B.

Figure 4 display histograms of pairwise similarities between samples for 6 similarity metrics for the Spider Realistic dev set, shown to illustrate the raw data leveraged by similarity aggregation methods. The distribution for sentence BERT cosine similarity indicates that generations typically tend to be semantically similar. Many of the similarities are 0 for the SQL-specific metric Makiyama because many generations cannot be parsed, in which case we default to 0. This issue was observed for other SQL metrics as well, making them less desirable for consistency-based UQ without some modifications. In contrast, the token/text metrics show some level of gradation and are able to capture more nuanced comparisons between generations, making it easier for our proposed aggregation methods to yield better calibrated confidences. Importantly, the results demonstrate that comparing a SQL with other generated SQLs, even if they happen to be syntactically invalid, is beneficial for the purpose of confidence estimation. We surmise that **standard token/text similarity metrics are suitable for consistency-based black-box UQ, even for outputs such as SQL**.

## 5.3 EFFECTIVENESS OF BLACK-BOX UQ FOR TEXT-TO-SQL

We investigate the effectiveness of black-box UQ by similarity aggregation using generations from 3 different models on the Spider dev set. We use hybrid sampling with 5 samples each over 6 temperatures (from 0.25 to 1.5 in increments of 0.25), with evaluations performed only on samples from the lower 3 temperatures since the higher temperatures provide generations with lower execution accuracy. This is done to mimic the realistic scenario where the user wishes to obtain confidence

estimates for only those samples they will even consider. Again, we split the data randomly into half for train/test sets, and repeat the experiment 5 times so as to study the variability of the results.

We compare our proposed approaches with 4 non black-box baselines: naive baselines that always return a score of either 0 or 1, a white-box non-verbalized approach that computes the avg. probability of tokens from logits, often used in question answering (Kuhn et al., 2022; Lin et al., 2024; Manakul et al., 2023), and a white-box verbalized approach from LLM self-evaluation (Kadavath et al., 2022). We also consider 2 baseline black-box aggregation methods as described previously (Lin et al., 2024). We do not consider approaches that use natural language inference for similarity (Kuhn et al., 2022; Chen & Mueller, 2024) or those requiring fine-tuning LLMs, since they are either unsuitable for SQL or require substantial training data. We use 2 of the best performing similarity metrics (Jaccard and Rouge-L) for all consistency-based methods and showcase 2 of our best proposed aggregations (Bayesian aggregation with Betas and classification with a random forest) for simplicity.

Table 2 presents results for 3 models on queries from the Spider dev set over 2 evaluation metrics – ACE and AUROC. Comparing the performance of each UQ method as shown in the rows, separately for each model, we observe that **the proposed black-box UQ methods consistently result in lower ACE as compared to baselines**. Classification with a random forest using the Rouge-L metric in particular is often highest performing. Yet again, **simple aggregation by averaging similarities performs reasonably well on AUROC**. Note that a comparison across models is inappropriate for this UQ evaluation as sampling results in different generations across models.

### 5.4 EFFECTIVENESS OF BLACK-BOX UQ FOR QUESTION ANSWERING

To analyze the generalizability of our proposed methods, we consider the open-book conversational question answering (QA) dataset CoQA (Reddy et al., 2019), the closed-book QA dataset Trivi-aQA (Joshi et al., 2017), as well as the more challenging closed-book QA dataset called Natural Questions (Kwiatkowski et al., 2019). We take the first 1000 questions from the corresponding dev sets for each dataset, and generate responses using two open-source models: **Granite 13B** (Mishra et al., 2024) and **LLaMA 2 70B** (Touvron et al., 2023).

We repeat the experiment around studying the effectiveness of black-box UQ by similarity aggregation from the previous sub-section. Tables 5 and 6 in Appendix C show results for generations from each data and model combination over 2 evaluation metrics, ACE and AUROC, respectively. We observe again that **the proposed black-box UQ methods, particularly aggregation by classification, consistently perform better on UQ metrics like ACE** as compared to baselines. This demonstrates that the methods may provide well-calibrated confidence estimates in other applications.

## 6 CONCLUSION

We present the first investigation around black-box UQ for the text-to-SQL task, where the consistency between a generation of interest and others is used as a proxy for our confidence in that generation. Specifically, we propose a general high-level similarity aggregation framework for UQ using pairwise similarities between multiple generated samples, as well as specific approaches within that framework. Our framework is quite general and can accommodate any similarity measure. We consider both text/token-level similarities such as the Jaccard coefficient and various ROUGE metrics as well as SQL specific similarities that exploit the clauses of the query. Through an extensive empirical evaluation using popular text-to-SQL benchmarks such as Spider and BIRD as well as state-of-the-art open-source code LLMs, we show that text/token-based similarity metrics such as Jaccard and Rouge-L are suitable for text-to-SQL UQ, and that the proposed similarity aggregation methods result in well-calibrated confidence estimates as measured by the adaptive calibration error. The proposed methods also generalize well to tasks such as question answering.

A **limitation** of some of our proposed methods is that they require a small training set where samples are generated in the same manner across training and testing. Also, although results show moderate gains over existing approaches on AUROC and AUARC, there is room for improvement when using these confidences to distinguish between correct and incorrect responses. A challenge with such consistency-based methods is that there are no guarantees for individual instances, and that their performance is task and model dependent. Further studies are needed to understand fundamental limitations of consistency-based UQ for generative tasks.

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

## A  EXPERIMENTAL DETAILS

**Prompt Templates.**  Table 3 shows prompt templates/examples for our few-shot approach for SQL generation with Codellama, as well as those for a baseline.

Table 3: Prompt templates for different tasks.

| Task | Prompt Template |
|------|-----------------|
| SQL Generation Codellama Few Shot | [INST]
Your task is to generate SQL query for the given question.
«SYS»
You are given the following database schema.
SQL query must have one or more of the tables and columns from the following schema.
If there is only one table in the generated query, alias should not be used. When there are multiple tables in the query, each should have alias like t1 and t2, and each column from those tables must use the alias. If column age is in table t1, and phone is in table t2, they should be written as t1.age and t2.phone.
SQL query must not contain more than one table unless required for the question. We want efficient queries.
Database: <database_name>
Tables:
Table <table_name> t1 <columns_names>
«/SYS»
Following are few shot examples of questions converted to SQL query.
question: Show the ids and names of all documents.
SQL query: SELECT document_id , document_name FROM Documents
question: Show the number of documents.
SQL query: SELECT count(*) FROM Documents
question: Find the name and access counts of all documents, in alphabetic order of the document name.
SQL query: SELECT document_name , access_count FROM documents ORDER BY document_name
question: Show all document ids and the number of paragraphs in each document. Order by document id.
SQL query:
[/INST] |
| p(True) Zero Shot | Instructions:
1. You are given an input question and a generated SQL query.  Your task is to check if the SQL query is correct with respect to the input question.
2. Your only output must be one of: True or False without any  lead-in, sign-off, new lines or any other formatting.
3. You are given the following database schema.
{}
True or False?
Input: {}
SQL query: {}
Output: |

**Baseline Details.**  We provide some additional details about baselines below:

- *spec-ecc* (Lin et al., 2024): We apply a threshold of 0.9 to keep only the selected eigen vectors for the spectral clustering with eccentricity baseline.
- *p-true* (Kadavath et al., 2022): We prompt the LLM used for generations to provide their belief about whether a generation is True or False. An illustration of the zero-shot prompt template is shown in Table 3.

Table 4: Comparing different similarity metrics and similarity aggregation approaches for consistency-based black-box UQ using generations from a few-shot codellama model on the BIRD dev set. See the caption of Table 1 for other experiment related details.

| Eval. Metric | ACE ↓ | | | | | AUROC ↑ | | | | |
|---|---|---|---|---|---|---|---|---|---|---|
| | spec-ecc | arith | beta | log-reg | rand-for | spec-ecc | arith | beta | log-reg | rand-for |
| Makiyama | 0.652 ±0.010 | 0.112 ±0.006 | 0.171 ±0.066 | 0.105 ±0.007 | 0.109 ±0.007 | 0.31 ±0.02 | 0.69 ±0.02 | 0.70 ±0.02 | 0.70 ±0.02 | 0.70 ±0.02 |
| Output type | 0.257 ±0.003 | 0.484 ±0.005 | 0.188 ±0.009 | 0.136 ±0.005 | 0.136 ±0.005 | 0.43 ±0.00 | 0.58 ±0.00 | 0.58 ±0.00 | 0.57 ±0.02 | 0.57 ±0.02 |
| Jaccard | 0.648 ±0.010 | 0.226 ±0.005 | 0.126 ±0.006 | 0.114 ±0.008 | 0.093 ±0.008 | 0.27 ±0.03 | 0.76 ±0.01 | 0.74 ±0.01 | 0.75 ±0.02 | **0.78** ±0.02 |
| Rouge1 | 0.460 ±0.011 | 0.388 ±0.005 | 0.137 ±0.008 | 0.097 ±0.007 | 0.090 ±0.006 | 0.31 ±0.01 | 0.73 ±0.01 | 0.77 ±0.01 | 0.77 ±0.01 | 0.77 ±0.01 |
| Rouge-L | 0.505 ±0.010 | 0.360 ±0.006 | 0.149 ±0.011 | 0.098 ±0.007 | **0.089** ±0.008 | 0.28 ±0.01 | 0.75 ±0.01 | 0.77 ±0.01 | 0.77 ±0.02 | 0.77 ±0.02 |
| Sbert-cos | 0.309 ±0.008 | 0.557 ±0.006 | 0.117 ±0.006 | 0.099 ±0.009 | 0.091 ±0.007 | 0.40 ±0.01 | 0.69 ±0.01 | **0.78** ±0.01 | 0.77 ±0.01 | 0.77 ±0.01 |

# B    EFFECT OF SIMILARITY METRIC AND AGGREGATION TECHNIQUE ON THE BIRD DATASET

We repeat the experiment around exploring the choice of similarity metric and similarity aggregation technique using generations from a few-show codellama model, using the BIRD dev set instead of the Spider Realistic dev set (as shown in Table 1 in the main text). Recall that temperature sampling is conducted over 6 temperatures and evaluations are performed using all 6 samples across all queries in the dataset. We split the data randomly into half for train/test sets, and repeat the experiment 5 times so as to study the variability of the results.

The rows in Table 4 correspond to 6 similarity metrics and the columns correspond to 5 UQ techniques with evaluations along 2 metrics – ACE and AUROC. For baselines, we consider two spectral clustering approaches for UQ that leverage a graph Laplacian matrix computed from pairwise similarities – one that uses eccentricity and another that uses degree (Lin et al., 2024); as mentioned previously, the latter is equivalent to simple aggregation using arithmetic mean.

Comparing similarity metrics, we observe again that all token/text-based metrics generally perform well on ACE with a powerful aggregation method such as a random forest classifier. Rouge-L and sbert-cos are high performing metrics for this dataset. We also note that our proposed aggregation methods are better for calibration metrics such as ACE rather than AUROC, as sometimes they only provide marginal improvements over averaging similarities with the arithmetic mean baseline.

# C    BLACK-BOX UQ FOR QUESTION ANSWERING DATASETS

To analyze the generalizability of our proposed methods, we consider the open-book conversational question answering (QA) dataset CoQA (Reddy et al., 2019), the closed-book QA dataset Trivi-aQA (Joshi et al., 2017), as well as the more challenging closed-book QA dataset called Natural Questions (Kwiatkowski et al., 2019). We take the first 1000 questions from the corresponding dev sets for each dataset. We generate responses using two open-source models: **Granite 13B** (Mishra et al., 2024) and **LLaMA 2 70B** (Touvron et al., 2023).

We repeat the experiment around studying the effectiveness of black-box UQ by similarity aggregation. Instead of the Spider dataset for text-to-SQL, we consider generations from the afore-mentioned 2 models on the 3 QA datasets. As before, we use hybrid sampling with 5 samples each over 6 temperatures (from 0.25 to 1.5 in increments of 0.25), with evaluations performed only on samples from the lower 3 temperatures since the higher temperatures provide generations with lower execution accuracy. We split the data randomly into half for train/test sets, and repeat the experiment 5 times so as to study the variability of the results. We use the same baselines as chosen previously, except the LLM self-evaluation approach (Kadavath et al., 2022) since prior studies show that other baselines outperform it on QA datasets (Manakul et al., 2023; Kuhn et al., 2022). We use the Jaccard and

Table 5: Comparing adaptive calibration error (ACE) across different UQ approaches for generations from 2 models each on 3 QA datasets. The non black-box (non BB) and black-box (BB) baselines are described in the main text. We consider 2 similarity metrics for the baseline and proposed black-box methods: Jaccard and Rouge-L.

| Dataset | | CoQA | | Natural Questions | | TriviaQA | |
|---|---|---|---|---|---|---|---|
| | | Llamma2-70B | Granite-13B | Llamma2-70B | Granite-13B | Llamma2-70B | Granite-13B |
| BB (baselines) | spec-ecc; jaccard | $0.346_{\pm 0.012}$ | $0.597_{\pm 0.006}$ | $0.396_{\pm 0.015}$ | $0.295_{\pm 0.013}$ | $0.664_{\pm 0.008}$ | $0.338_{\pm 0.017}$ |
| | spec-ecc; rouge-L | $0.226_{\pm 0.007}$ | $0.579_{\pm 0.009}$ | $0.392_{\pm 0.011}$ | $0.285_{\pm 0.008}$ | $0.693_{\pm 0.009}$ | $0.351_{\pm 0.015}$ |
| | arith; jaccard | $0.233_{\pm 0.006}$ | $0.056_{\pm 0.016}$ | $0.043_{\pm 0.006}$ | $0.314_{\pm 0.014}$ | $0.086_{\pm 0.009}$ | $0.258_{\pm 0.012}$ |
| | arith; rouge-L | $0.405_{\pm 0.007}$ | $0.119_{\pm 0.008}$ | $0.132_{\pm 0.011}$ | $0.355_{\pm 0.015}$ | $0.042_{\pm 0.008}$ | $0.0308_{\pm 0.013}$ |
| BB (proposed) | bayes-beta; jaccard | $0.266_{\pm 0.022}$ | $0.182_{\pm 0.012}$ | $0.384_{\pm 0.016}$ | $0.252_{\pm 0.015}$ | $0.114_{\pm 0.015}$ | $0.241_{\pm 0.088}$ |
| | bayes-beta; rouge-L | $0.395_{\pm 0.007}$ | $0.172_{\pm 0.013}$ | $0.273_{\pm 0.012}$ | $0.242_{\pm 0.019}$ | $0.092_{\pm 0.012}$ | $0.257_{\pm 0.114}$ |
| | clf-rf; jaccard | $0.034_{\pm 0.007}$ | $0.045_{\pm 0.012}$ | $0.043_{\pm 0.010}$ | $\mathbf{0.050}_{\pm 0.012}$ | $0.033_{\pm 0.009}$ | $0.059_{\pm 0.015}$ |
| | clf-rf; rouge-L | $\mathbf{0.031}_{\pm 0.009}$ | $\mathbf{0.041}_{\pm 0.016}$ | $\mathbf{0.039}_{\pm 0.010}$ | $0.052_{\pm 0.020}$ | $\mathbf{0.030}_{\pm 0.007}$ | $\mathbf{0.055}_{\pm 0.018}$ |

Table 6: Comparing AUROC across different UQ approaches for generations from 2 models each on 3 QA datasets. The non black-box (non BB) and black-box (BB) baselines are described in the main text. We consider 2 similarity metrics for the baseline and proposed black-box methods: Jaccard and Rouge-L.

| Dataset | | CoQA | | Natural Questions | | TriviaQA | |
|---|---|---|---|---|---|---|---|
| | | Llamma2-70B | Granite-13B | Llamma2-70B | Granite-13B | Llamma2-70B | Granite-13B |
| Non BB (baselines) | always 0 | $0.50_{\pm 0.00}$ | $0.50_{\pm 0.00}$ | $0.50_{\pm 0.00}$ | $0.50_{\pm 0.00}$ | $0.50_{\pm 0.00}$ | $0.50_{\pm 0.00}$ |
| | always 1 | $0.50_{\pm 0.00}$ | $0.50_{\pm 0.00}$ | $0.50_{\pm 0.00}$ | $0.50_{\pm 0.00}$ | $0.50_{\pm 0.00}$ | $0.50_{\pm 0.00}$ |
| | avg. prob | $0.54_{\pm 0.04}$ | $0.73_{\pm 0.01}$ | $0.72_{\pm 0.02}$ | $0.67_{\pm 0.02}$ | $0.79_{\pm 0.01}$ | $0.65_{\pm 0.02}$ |
| BB (baselines) | spec-ecc; jaccard | $0.26_{\pm 0.04}$ | $0.17_{\pm 0.01}$ | $0.25_{\pm 0.02}$ | $0.25_{\pm 0.01}$ | $0.13_{\pm 0.01}$ | $0.24_{\pm 0.02}$ |
| | spec-ecc; rouge-L | $0.27_{\pm 0.03}$ | $0.17_{\pm 0.02}$ | $0.22_{\pm 0.01}$ | $0.24_{\pm 0.01}$ | $0.11_{\pm 0.01}$ | $0.21_{\pm 0.02}$ |
| | arith; jaccard | $0.74_{\pm 0.04}$ | $0.83_{\pm 0.01}$ | $0.76_{\pm 0.02}$ | $0.76_{\pm 0.01}$ | $0.88_{\pm 0.01}$ | $0.77_{\pm 0.02}$ |
| | arith; rouge-L | $0.75_{\pm 0.03}$ | $0.84_{\pm 0.01}$ | $0.79_{\pm 0.02}$ | $\mathbf{0.77}_{\pm 0.01}$ | $\mathbf{0.91}_{\pm 0.02}$ | $\mathbf{0.81}_{\pm 0.02}$ |
| BB (proposed) | bayes-beta; jaccard | $0.74_{\pm 0.03}$ | $0.82_{\pm 0.02}$ | $0.76_{\pm 0.02}$ | $0.75_{\pm 0.01}$ | $0.88_{\pm 0.02}$ | $0.77_{\pm 0.02}$ |
| | bayes-beta; rouge-L | $0.75_{\pm 0.03}$ | $0.82_{\pm 0.03}$ | $\mathbf{0.79}_{\pm 0.02}$ | $\mathbf{0.77}_{\pm 0.01}$ | $\mathbf{0.91}_{\pm 0.02}$ | $\mathbf{0.81}_{\pm 0.02}$ |
| | clf-rf; jaccard | $0.84_{\pm 0.02}$ | $0.85_{\pm 0.01}$ | $0.75_{\pm 0.01}$ | $0.72_{\pm 0.02}$ | $0.86_{\pm 0.02}$ | $0.76_{\pm 0.01}$ |
| | clf-rf; rouge-L | $\mathbf{0.87}_{\pm 0.01}$ | $\mathbf{0.87}_{\pm 0.01}$ | $0.78_{\pm 0.01}$ | $0.73_{\pm 0.02}$ | $0.89_{\pm 0.03}$ | $0.78_{\pm 0.01}$ |

Rouge-L similarity metrics for all consistency-based methods and showcase 2 of our best proposed aggregations (Bayesian aggregation with Betas and classification with a random forest).

Tables 5 and 6 present results for generations from each data and model combination over 2 evaluation metrics, ACE and AUROC, respectively. Comparing the performance of each UQ method as shown in the rows, separately for each model, we observe again that the proposed black-box UQ methods consistently result in lower ACE as compared to baselines. The proposed methods are often highest performing, even for QA datasets. Classification using random forests performs best for ACE, and simple aggregation by averaging similarities based on Rouge-L performs reasonably well on AUROC.

