# OpenReview forum: "Consistency-based Black-box Uncertainty Quantification for Text-to-SQL by Similarity Aggregation"
_ICLR.cc/2025/Conference — ICLR 2025 Conference Withdrawn Submission_

### Official Review · Reviewer_pTge · 2024-10-22

**Soundness:** 2
**Presentation:** 3
**Contribution:** 2
**Rating:** 3
**Confidence:** 3

**Summary:**

This paper explores multiple approaches to consistency-based black-box (no access to the internal state of the model) uncertainty quantification (UQ) for text-to-SQL and proposes a new high-level non-verbalized similarity aggregation method. The experiments show that the proposed method outperforms other baselines in model calibration for both text-to-SQL and question answering datasets.

**Strengths:**

- The motivation of the paper is convincing.
- The proposed method shows consistent improvements in the designed setup.

**Weaknesses:**

- The scope of this work seems a bit small, which limits the impact of the paper. I definitely agree on the importance of the text-to-SQL task and that calibration is important. However, there are many black-box methods (as mentioned in the related works), and focusing only on the particular set of consistency-based methods limits the impact.
- Although the proposed method performs well in the designed setup, it would be much more impactful to include comparisons with other UQ methods, such as conformal prediction, which are closely related to consistency-based black-box methods [1, 2].

[1] Quach, V., Fisch, A., Schuster, T., Yala, A., Sohn, J. H., Jaakkola, T. S., & Barzilay, R. Conformal language modeling. ICLR 2024.
[2] Su, J., Luo, J., Wang, H., & Cheng, L. (2024). Api is enough: Conformal prediction for large language models without logit-access. ArXiv 2024

**Questions:**

- Open-source models allow users to access their internal states, while API-based LLMs do not. However, exploring open-source models for black-box approaches seems a bit self-contradictory.
- Why is a verbalized method introduced as white-box? (Line 186). Some verbalization methods do not require access to logits (e.g., qualifying phrases).
- In terms of paper organization, if the proposed method works well for question-answering datasets (also, are they all long-sequence generation tasks?), I'm curious why the authors titled the paper UQ for text-to-SQL. Is there a specific reason behind this decision?

---

> ### Author Response · Authors · 2024-11-21
> **Response**
>
> We thank the reviewer for their valuable feedback and suggestions. We provide responses to specific questions and comments below.
>
> > **Re: scope:**
>
> We clarify that the scope of our work is broad and that the proposed approach is applicable to numerous tasks. Although UQ for text-to-SQL is our main motivation, we showed generalizability to tasks such as QA in Section 5.4. Also, perhaps there is some misunderstanding about the baselines, but we have included a broad suite of those including verbalized and white-box approaches (not just consistency-based approaches). We would appreciate any suggestions that the reviewer may have to further highlight breadth of our work.
>
> > **Re: conformal prediction references:**
>
> We thank the reviewer for the references. While conformal prediction for LLMs is interesting and clearly related to uncertainty quantification/estimation, the goal is different in that it aims to find a prediction set that contains the desired result with confidence guarantees. The target tasks in the suggested papers are for QA or summarization, where algorithms generate multiple answers to cover the correct answer. The evaluation metrics for conformal prediction are the coverage rate and the prediction set size. In contrast, the goal of our paper is to find a well-calibrated confidence estimate that reduces the calibration error. The two suggested papers cannot be suitable baselines fir our purposes.
>
> ***Responses to Questions***
>
> > **”Open-source models allow users to access their internal states, while API-based LLMs do not. However, exploring open-source models for black-box approaches seems a bit self-contradictory.”**
>
> The reviewer is correct that API-based LLMs are often the ones without access to internal states. Note that our methods apply just as much to API-based LLMs, but our experiments were restricted to open-source LLMs for practical reasons.
>
> > **”Why is a verbalized method introduced as white-box? (Line 186). Some verbalization methods do not require access to logits (e.g., qualifying phrases).”**
>
> We agree that some verbalized methods are not white-box, as the reviewer indicates, but the one mentioned in this line happens to be white-box as it needs the logit for the True or False output. We will clarify this in our revision.
>
> > **”In terms of paper organization, if the proposed method works well for question-answering datasets (also, are they all long-sequence generation tasks?), I'm curious why the authors titled the paper UQ for text-to-SQL. Is there a specific reason behind this decision?”**
>
> We chose to focus our paper on estimating well-calibrated confidences for the text-to-SQL task, as this is the main motivation behind the work. This choice requires specifying information such as related work, choice of metrics, datasets, etc. Section 5.4 is intended to show generalizability of the work to QA (which is the sole focus of some related work) and show broad scope of the proposed approach. Also, to clarify, some of the QA datasets involve short generation tasks.
>
> We hope that we addressed the reviewer’s main concerns, particularly about scope of the work, which seems to be a major factor in the reviewer’s rating. If this is indeed the case, we request them to consider increasing their rating.

---

### Official Review · Reviewer_ovi8 · 2024-10-28

**Soundness:** 3
**Presentation:** 2
**Contribution:** 3
**Rating:** 5
**Confidence:** 3

**Summary:**

The paper explores LLMs can understand their own limitations through uncertainty quantification (UQ) on text-to-SQL task, where LLMs convert natural language queries into structured queries for database insights. This paper propose a new similarity aggregation approach for estimating confidence in complex tasks, along with techniques to train confidence models using limited data. Extensive experiments across various text-to-SQL datasets reveal that their proposed methods improve confidence estimation compared to existing baselines, although there is still potential for enhancing performance in tasks like selective generation.

**Strengths:**

* Focus on the Uncertainty quantification  in  text-to-SQL task, which is important aspect in practice.
* This paper propose a consistency-based method to examines the effectiveness of Black-box UQ methods by using the consistency of generated outputs as a confidence measure.

**Weaknesses:**

* In line 77, the author points out: "The implicit underlying assumption behind consistency-based approaches is that when a generated response is more different from others, it is more likely to be incorrect, implying that responses that are consistently similar are more likely to be correct." Is there any evidence for this? Why does a response being more different from others indicate incorrectness rather than uncertainty?
* In Figure 1, parts (a) and (c) show that the correct response is closer to others, but this is not very clear in part (b). Providing examples to illustrate the data format would be helpful.
* The ACE metric is calculated by: \text{acc}(b, k) - c(b, k)acc(b,k)−c(b,k). A lower ACE means either low accuracy or high confidence error. It would be beneficial to provide more details about this calculation.
* The proposed similarity method can evaluate if the model knows these inputs, but how does it perform with data the model hasn't seen?
* The results in Table 1 and Table 2 do not clearly show the strengths of the proposed method. It would be better to highlight the significance of each value in bold.

**Questions:**

* AUROC  and AUARC could shown the accuracy  of the model on  text-to-SQL task?
* The hyper-parameter m in line 233 and B in line 349  is same?
* Some setting like "bayes-beta" is not clear it's meaning.

---

> ### Author Response · Authors · 2024-11-21
> **Response**
>
> We thank the reviewer for their valuable feedback and suggestions. We provide responses to specific questions and comments below.
>
> > **Re: evidence about implicit underlying assumption behind consistency-based approaches:**
>
> We have indeed observed empirical evidence for the underlying assumption about consistency; for instance, the mean similarity between correct generations is less than that between incorrect generations across datasets. Fig. 1 is a visualization of the assumption, although it is qualitative and only covers a few queries. We stress that the results of the work provide evidence for the assumption, since they would not generally hold if the assumption did not hold at least to some degree.
>
> > **Re: Fig. 1(b):**
>
> We agree with the reviewer’s observation about Fig. 1(b). However, we note that this instance indicates importantly that the consistency assumption does not hold the same way for all instances — it is a statistical assumption, i.e. holds generally but not to all instances. However, we appreciate the point about showing examples and will consider adding some in the Appendix.
>
> > **Re: ACE metric computation:”**
>
> We provided a short description of the ACE metric computation in line 346. Space-permitting, we will add more detail here.
>
> > **Re: results in Tables 1 and 2:**
>
> We have already used bold font to highlight the best results in both tables. Was the bold font hard to identify? We will see how we can make the results clearer.
>
> > **Re: comment: “The proposed similarity method can evaluate if the model knows these inputs, but how does it perform with data the model hasn't seen?”**
>
> We are not sure we understand this comment correctly — we clarify that the approach is intended to work for unseen instances. For the low data methods, we assume that the nature of sampling is the same across training and test sets, but that’s the only assumption being made.
>
> ***Responses to Questions***
>
> > **”AUROC and AUARC could shown the accuracy of the model on text-to-SQL task?”**
>
> We clarify that it is not our intent to improve execution accuracy of the underlying task but to estimate well-calibrated confidences. AUROC gauges how well one can distinguish between correct and incorrect generations using confidences, and AUARC gauges how performance changes as confidence is used to determine which instances to abstain from answering.
>
> > **”The hyper-parameter m in line 233 and B in line 349 is same?:**
>
> No, they are different — m is the number of generations/samples and B is the number of bins for computing adaptive calibration error (ACE).
>
> > **Some setting like "bayes-beta" is not clear it's meaning.**
>
> Thanks for pointing this out — we will try to make it clearer. Bayes-beta refers to the proposed approach of Bayesian aggregation with Beta distributions.
>
> We hope that we have addressed the reviewer’s main concerns, and if this is indeed the case, we request them to consider increasing their rating.

---

### Official Review · Reviewer_DQgh · 2024-10-30

**Soundness:** 3
**Presentation:** 3
**Contribution:** 2
**Rating:** 5
**Confidence:** 4

**Summary:**

This paper investigates the effectiveness of a black-box Uncertainty Quantification (UQ) technique for text-to-SQL, where the confidence of the generated SQL statement is computed by the set of generated SQL statements.  More specifically, given a question, the proposed method first generates three kinds of SQL statements based on different sampling mechanisms facilitating temperatures in LLMs.   A confusion matrix of pair-wise similarity of the generated SQL statements is computed by different metrics.  Three kinds of aggregation methods are applied to compute the confidence of the generated SQL statement.  Empirical evaluations on three text-to-SQL datasets and various settings have been conducted to verify the advantage of the UQ technique.

**Strengths:**

1. The paper identifies an interesting research direction to evaluate the confidence score of SQL generated by the model in a black-box manner.
2. The paper proposes a similarity aggregation framework for estimating confidence, in which a fusion of various methods are attempted.

**Weaknesses:**

1. About the technical contribution of the work.  The learning procedure mainly lies in the confusion matrix to aggregation results.  This seems common in the field.  Though it can provide a confidence score, it seems not tackled to improve the performance of text-to-SQL.
2. The correctness of Fig. 1 is doubtful. The caption states there are 30 generations, but the figure shows 35 generations. To address this discrepancy, it is recommended that the authors clarify the difference between the number of generations mentioned in the caption and those presented in the figure. Additionally, the authors should explicitly state whether overlapping generations are allowed and explain how such overlaps are handled in the figure.
3. The experimental comparison seems not sufficient.  For example,
3.1) The paper does not compare the experiments with sufficiently strong baselines. The current comparison on the non-black-box baseline includes only a few simple cases. It is recommended that the authors incorporate more robust baselines, such as Semantic Entropy (SE)[1] as a non-black-box baseline, to strengthen the experimental evaluation.
[1] Semantic uncertainty: Linguistic invariances for uncertainty estimation in natural language generation.  https://arxiv.org/abs/2302.09664
3.2) The paper does not test the effect of the number of generations.  The paper should provide more comparison results to show the potential impact factors of the proposed method.

**Questions:**

1. When sampling, does repeated generation occur? If so, how do the authors handle these repeated generations, and are they retained? It would be helpful for the authors to clarify this and provide more comparison results in the paper, as repeated generations may impact the weight of the confusion matrix and potentially affect the overall results.
2. See the questions in Weakness.

---

> ### Author Response · Authors · 2024-11-20
> **Response**
>
> We thank the reviewer for their valuable feedback and suggestions. We provide responses to specific questions and comments below.
>
> > **Re: improving text-to-SQL performance:**
>
> Note that it is not our intent to improve performance of the underlying task. We are interested in estimating well-calibrated confidence estimates, just like other work on UQ. This is the main objective of our proposed UQ module (for a text-to-SQL system).
>
> > **Re: Fig 1:**
>
> We thank the reviewer for spotting this discrepancy. This figure has 35 samples per query but the experiments use 30 samples per query. The figure itself is correct but the caption needs to be changed. We will either modify the plots or the caption accordingly.
>
> Identical outputs are indeed allowed and are handled the same as any other output pair, where pairwise similarity becomes 0 for such outputs. Fig. 1 is just a visualization to convey the consistency assumption that is implicit in consistency-based approaches, so such outputs would be overlapping in the visualization.
>
> > **Re: proposed baseline and effect of generations:**
>
> We make an important clarification — the Kuhn et al. 2022 work on semantic entropy is NOT a suitable baseline for our work. Their use of natural language inference is crucial to their method and this does not make sense for SQL outputs — what does it mean for a SQL query to entail another one? We have cited this paper several times and even mentioned why it is not suitable for text-to-SQL in line 491. Note also that Lin et al. 2024 has shown superior performance over Kuhn et al. 2022 on QA tasks, and we use that as a baseline in our work.
>
> We thank the reviewer for the excellent suggestion of ablation on number of generations and will add this in the Appendix.
>
> > **Re: repeated generations:**
>
> Yes, repeated generations are indeed allowed — please see our answer above about overlapping outputs.
>
> We believe we have addressed most of the reviewer’s concerns, including unsuitability of a proposed baseline, fixing a caption error in Figure 1, and clarifying that repeated generations are allowed. If they agree, we hope they will consider increasing their rating.

---

> > ### Comment · Reviewer_DQgh · 2024-11-25
> >
> > Thanks to the authors for the responses. I will maintain the original rating since I do not see much update.

---

> > > ### Author Response · Authors · 2024-11-27
> > > **Revised version uploaded**
> > >
> > > We have posted a revised version of the paper, based on some of the feedback. We have not yet incorporated the reviewer's suggestion around ablation on number of generations due to time limitations but will plan to add that later. We hope we have provided sufficient clarifications in our previous response.

---

### Official Review · Reviewer_GU7m · 2024-11-04

**Soundness:** 2
**Presentation:** 3
**Contribution:** 2
**Rating:** 3
**Confidence:** 4

**Summary:**

Uncertainty quantification is important for generative tasks like text-to-SQL. Black-box UQ has many advantages like robustness to system changes. This paper investigate black-box UQ for text-to-SQL, where the consistency between a generated output and other sampled generations is used as a proxy for estimating its confidence. It proposes a high-level non-verbalized  similarity aggregation approach for complex generative tasks.

**Strengths:**

None

**Weaknesses:**

- The experiments are not solid. The metric (ROUGE) used to evaluate the method could only reflect the syntactic similarity but hard to evaluate semantic-level correlation. There is only 1 semantic-level metric (sentence bert) to evaluate the method. However, 1) even semantic similarity metrics couldn’t reflect “contradiction”, e.g., the similarity is still high but the semantic meanings are contradictory. 2) it seems for sentence bert, SQL is out-of-distribution. The similarity measurement may not be reliable. Therefore, the results obtained from those experiments are possibly meaningless and the conclusion of this paper is ungrounded.
- The motivation behind this work is unclear, and its scope appears limited. It focuses narrowly on a single task (text-to-SQL) and uncertainty quantification specific to that task. Broader and more robust results would be valuable to support practical applications of this work.
- In line 46-75, there are a few instances of unclear phrasing and awkward language use that may hinder comprehension.
- It’s unclear that how this method tackles “over confidence” from LLMs. It’s possible that LLM has a false belief and it doesn’t matter with how the LLM evaluate itself (line 94: ”when asking LLMs for probabilities). It’s strange that the paper mentions so. What if the model generates very similar text samples but all of them are wrong? Could the proposed similarity based method handle this case?

**Questions:**

None

---

> ### Author Response · Authors · 2024-11-20
> **Response**
>
> We thank the reviewer for their feedback. It appears though that they have misunderstood several aspects of the paper, so we clarify matters by responding to specific questions and comments below.
>
> > **Re: the following comments about metrics:** “The metric (ROUGE) used to evaluate the method could only reflect the syntactic similarity but hard to evaluate semantic-level correlation. There is only 1 semantic-level metric (sentence bert) to evaluate the method … The similarity measurement may not be reliable. Therefore, the results obtained from those experiments are possibly meaningless and the conclusion of this paper is ungrounded.”
>
> Firstly, the phrase “metric (ROUGE) used to evaluate the method” is factually incorrect — the metrics used for evaluation are ACE, AUROC, and AUARC. We believe the reviewer is referring instead to the choice of similarity metric. They seem to have however completely missed that 1) the output (for most of the paper) is SQL, and 2) we have used numerous similarity measures (see Section 3.2 on page 4) spanning syntactic, semantic, and SQL-specific metrics. Various experiments discuss the implications of the choice of similarity metric. We have even explained our conjecture for why syntactic similarity measures such as ROUGE may be doing well for UQ for text-to-SQL starting around line 463. The scope of the experiments is much broader than the comment suggests and the conclusions are very much meaningful, given the extensive nature of the empirical study.
>
> > **Re: motivation and scope:**
>
> The motivation for the work is clearly stated in Section 1. Text-to-SQL is a hugely consequential task and widely studied in the AI and data communities. We chose to focus on the task as it has complexities that go beyond QA tasks. Section 5.4 even discusses the generality of the ideas by conducting experiments on QA datasets. The scope of our work is in fact broader than numerous papers published previously on QA tasks alone, since we experiment with more datasets (see Kuhn et al. 2022 and Lin et al. 2024 for comparison).
>
> > **Re: awkward language in lines 46-75:**
>
> Could the reviewer please specify the lines that are unclear? This paragraph seems clear to us and is important to appreciate the motivation of the work. Specific feedback will help us in our revision.
>
> > **Re: comments about verbalization and consistency:** “It’s unclear that how this method tackles “over confidence” from LLMs. It’s possible that LLM has a false belief and it doesn’t matter with how the LLM evaluate itself (line 94: ”when asking LLMs for probabilities). It’s strange that the paper mentions so. What if the model generates very similar text samples but all of them are wrong? Could the proposed similarity based method handle this case?”
>
> We address yet another misunderstanding — our method does NOT ask LLMs to provide confidences; it is a non-verbalized method and merely uses LLM outputs. Note that consistency-based approaches all make assumptions about consistency and the correctness of LLM generations. If this is not the case and an LLM consistently provides highly similar incorrect answers, then this will not be accounted for by ANY consistency-based method, including ours. No method is perfect but we have empirically shown that our proposed method performs better than relevant baselines, particularly for the ACE evaluation metric.
>
> We hope our responses have clarified misunderstandings and request the reviewer to completely re-evaluate the work and update their rating accordingly.

---

> > ### Comment · Reviewer_GU7m · 2024-11-21
> > **Re: Response**
> >
> > Thanks the authors for the response.
> >
> > - Metric
> >     - I was referring to the use of the metric (ROUGE) as a similarity estimation metric for computing confidence and its inclusion in the experiments. The authors have indeed correctly understood my concerns.
> >         - However, the output format (SQL) does not directly imply the usage of ROUGE. As the paper mentioned, they have considered some specific SQL metrics (e.g., Makiyama). But the choice of ROUGE or cosine similarity of Sentence BERT is still not justified. The authors claim that they later found the evaluation to be effective with ROUGE in the experiment, which comes across as a post-hoc explanation. The rationale for selecting these metrics should be better articulated prior to conducting the experiment.
> >         - In line 471, the paper mentions that “many generations cannot be parsed”, which possibly explains why, in Figure 4 “Makiyama”, the distribution seems to be very different from those of syntactic or semantic metrics for natural language. Since many of the outputs from LLMs are not parsable SQL sequences, the experimental results based on ROUGE or Sentence BERT are not reliable because they don’t care if the SQL query is valid or not. In this case, the problem falls back into a more general form (text sequence) and the authors may want to re-think that experimenting on non-SQL sequence could validate the claim made by the paper (Given the topic of the paper is UQ for text-to-SQL).
> > - Regarding language usage in line 046-075
> >     - line 050: “…, as gauged by the degree to which they match the empirical accuracy for that prediction” seems cumbersome. Something like “gauged by how closely they align with the empirical accuracy of the predictions.” might be better.
> >     - line 052-053: “…, where one only assumes access to the generating model under consideration (such as an LLM), …” seems awkward. Maybe the authors want to simplify it as “assuming access only to the model being used (such as an LLM)”. Also, I think what the authors really want to highlight here is that the user can only get access to the “generated text or sequence” from an LLM. But the original text seems to be obscure.
> >     - line 074: “… an open question around how such methods will fare while estimating confidence for more complex generative tasks …” could be more concise. For example, “an open question remains about how these methods perform in estimating confidence for…”
> > - Overconfidence
> >     - I was NOT saying that this method involves asking LLMs to provide confidence. Please read my comment carefully. My concern was how this method tackles “overconfidence” as the paper itself mentions in line 093 - 094 (“**avoiding** some empirically observed concerns around potential **overconfidence** when asking LLMs for probabilities”). Additionally, in Figure 1 (b), there is a clear example where incorrect generations are also clustered together, which poses a risk of overconfidence by LLMs. Note that the paper is based on the assumption that “when a generated response is more different from others, it is more likely to be incorrect, implying that responses that are consistently similar are more likely to be correct” (line 078-080), it is necessary to justify that consistency-based approach works for this specific task (text-to-SQL) and not negatively affected by LLM overconfidence.
> >
> >
> > - Some of the concerns are addressed by the authors. I will update the scores accordingly.

---

> > > ### Author Response · Authors · 2024-11-22
> > >
> > > We thank the reviewer for their additional response and willingness to re-assess the work. Their clarifications are very useful for us to provide further justifications and explanations as well as revise the draft. Our responses follow:
> > >
> > > > **Re: metric:**
> > >
> > > It may help to appreciate that it is not necessary to use only valid SQLs to estimate confidence for any particular SQL. In fact, the results demonstrate that comparing with generated invalid SQLs is beneficial for the purpose of confidence estimation. All generations are sequences of tokens after all, and one could potentially use any similarity metric to compare generations. We wanted to choose a wide enough range of suitable similarity metrics and observe empirical results, where we identified that some of the syntactic metrics work better for confidence estimation. It was not obvious to us before that this would be the case for text-to-SQL, so this is a finding from the work. While experimenting with QA datasets, we also consider various similarity metrics (except the ones specific to SQL output of course), and again syntactic ones seem to perform the best for confidence estimation. We will clarify this overall insight in our revision.
> > >
> > > > **Re: language usage:**
> > >
> > > We thank the reviewer for these specific suggestions, which will help make this paragraph clearer and more concise.
> > >
> > > > **Re: Fig. 1(b) and overconfidence:”**
> > >
> > > We agree with the reviewer that Fig, 1(b) highlights a situation where it is tricky to distinguish between correct and incorrect instances. The underlying assumption about consistency is a statistical one and does not apply to all instances — this seems to be at the heart of the misunderstanding. Note that any consistency-based approach is unlikely to provide a good confidence estimate in such a situation. And yet consistency-based approaches perform well over a test set for different tasks and datasets, better than strong baselines.
> > >
> > > We appreciate the discussion and hope our responses have provided further clarifications.

---

### Note · Authors · 2024-12-15

I have read and agree with the venue's withdrawal policy on behalf of myself and my co-authors.